# OpenReview forum: "Toward Efficient Kernel-Based Solvers for Nonlinear PDEs"
_ICLR.cc/2025/Conference — Submitted to ICLR 2025_

### Official Review · Reviewer_vnHZ · 2024-10-29

**Soundness:** 2
**Presentation:** 2
**Contribution:** 1
**Rating:** 1
**Confidence:** 4

**Summary:**

This work considers a slightly different way to solve PDE via the kernel-ridge-regression.  To put this work in context, see below.

**Strengths:**

This work considers a slightly different way to solve PDE via the kernel-ridge-regression. See the assessment below.

**Weaknesses:**

To put this work in the context of RBF (e.g., [2,3]), let's consider solving a simple PDE problem,
$$
\\mathcal{P}(u) := -u'' + u = f, \\quad (1) %\\label{PDE}
$$
on a one-dimensional periodic domain (ignoring the boundary condition for simplicity). Using the notation in the paper, suppose we let the solution be
$$
u(x) = k(x,\mathcal{M}) \alpha, \quad (2)%\\label{ansatz}
$$
where $$\\mathcal{M} = \\{\mathbf{x}\_1,\ldots,\mathbf{x}\_M\\}$$ and $k$ denotes any kernel (which is positive definite and it corresponds to the RKHS space $\mathcal{H}$), which norm is defined as,
$$
\| u \|^2_{\mathcal{H}} = \alpha \mathbf{K} \alpha,
$$

where $\mathbf{K}= k(\mathcal{M},\mathcal{M})\in \mathbb{R}^{M\times M}$ is the Gram matrix of $k$.

Inserting the ansatz in (2) into the PDE in (1), one can deduce the following linear system
$$
\mathbf{A}\alpha := (-\mathbf{K}''+c\mathbf{K}) \alpha = \mathbf{f}.
$$
where $\mathbf{K}''$ is a shorthand notation for the Gram matrix corresponds to $k"$ and $\mathbf{f} = (f(\mathbf{x}\_1),\ldots,f(\mathbf{x}\_M))$. At this point, let's just ignore the computational feasibility. Since $\mathbf{A}$ is invertible, one can simply solve this $M\times M$ linear problem. One can write the solution as $u(x) = k(x,\mathcal{M})\mathbf{A}^{-1}\mathbf{f}.$  I am not sure why solving this problem severely worse than the proposed method if the problem is invertible as noted in the last paragraph in Section 5. Many convergence results have been reported in literature (see [2,3] and the references therein).

Now, let's look at the Kernel Ridge Regression for this PDE problem, that is, we solve:
$$
\\min_{u\\in \\mathcal{H}}  \\frac{1}{M}\\sum_{i=1}^M (\\mathcal{P}(u)(\\mathbf{x}\_i) - f\_i)^2 +  \\lambda \\|u\\|^2_{\\mathcal{H}},
$$
Inserting the ansatz in in (2), we rewrite this optimization problem as,
$$
\min_\alpha \frac{1}{M} (\mathbf{A}\alpha - \mathbf{f})^\top (\mathbf{A}\alpha - \mathbf{f}) + \lambda \alpha \mathbf{K}\alpha. (2a)
$$
Taking derivative and set the equation to zero, we arrive at solving the following linear problem,
$$
(\mathbf{A}^2 + M\lambda \mathbf{K}) \alpha = \mathbf{A} f, \quad (3)%\label{linearproblem1}
$$
using the fact that $\mathbf{A}$ is symmetric. If this is invertible, one can simply write the solution as,
$$
u(x) = k(x,\mathcal{M})(\mathbf{A}^2 + M\lambda \mathbf{K})^{-1}\mathbf{A} f.\quad (4) %\label{sol1}
$$
The key idea in this paper is to consider the solution of the following form,
$$
u(x) = k(x,\mathcal{M}) \mathbf{K}^{-1}\eta.\quad (5)%\label{ansatz2}
$$
In such case, the minimization problem becomes,
$$
\min_\eta \frac{1}{M} (\mathbf{A}\mathbf{K}^{-1}\eta - \mathbf{f})^\top (\mathbf{A}\mathbf{K}^{-1}\eta - \mathbf{f}) + \lambda \eta \mathbf{K}^{-1}\eta,\quad (6)
$$
and following the standard calculus, the solution satisfies,
$$
(\mathbf{K}^{-1}\mathbf{A}^2\mathbf{K}^{-1} + M \lambda \mathbf{K}^{-1}) \eta =\mathbf{K}^{-1} \mathbf{A}\mathbf{f},
$$
which is equivalent to multiplying (3) from the left by $\mathbf{K}^{-1}$ and letting $\alpha = \mathbf{K}^{-1}\eta$. If we solve this problem, we end up with
$$
u(x) = k(x,\mathcal{M})\mathbf{K}^{-1}(\mathbf{K}^{-1}\mathbf{A}^2\mathbf{K}^{-1} + M\lambda \mathbf{K}^{-1})^{-1}\mathbf{K}^{-1} \mathbf{A} f, %\quad (7)\label{soln2}
$$
which I believe is identical to (4) if $\mathbf{K}$ is invertible. In order to write the proposed ansatz in (5), $\mathbf{K}$ is assumed to be invertible. In fact, the author considers speeding up the inversion of $\mathbf{K}$ using an existing method in literature as reported in p.4.

While I fully understand that the authors consider a minimization algorithm to solve this problem, I just cannot see any new idea with this formulation. BTW, if the PDE is nonlinear, I believe RBF will result in a minimization problem as well.

Beyond this issue, I also do not understand why the choice of tensorial kernel makes any sense except for computational convenience. Basically, the proposed idea in Eq.(9) in the manuscript is to consider the tensor product of kernels that compares scalars (a component of the data). So, the solutions are chosen in the space of functions induced by a kernel that only compares distances in one dimension. The classical approach is to choose a kernel that compares distances in $\mathbb{R}^d$. Mathematically, it is unclear why the proposed choice should always be a better choice aside from the numerical advantage.

Finally, the numerical simulations are not convincing. I am not sure what DAKS is. It is not surprising the scheme is more accurate than PINN as it is well known that neural-network solutions are not accurate anyway. Lastly, the fact that this scheme beats the finite-difference scheme is also not so surprising based on the classical Numerical Analysis understanding in approximation of derivatives. One can check also the following paper that reported the advantage of RBF over finite-difference [1].

Based on these (comparisons with well-established classical literature), I am not sure this paper merits consideration for publication in ICLR.

References.
[1] B. Fornberg. The pseudospectral method: Comparisons with finite differences for the elastic wave equation.
 Geophysics, 52(4):483--501, 1987.

[2] B. Fornberg and N. Flyer.  A primer on radial basis functions with applications to the geosciences.
SIAM, 2015.

[3] B. Fornberg and N. Flyer. Solving pdes with radial basis functions. Acta Numerica, 24:215--258, 2015.

**Questions:**

Based on the comments above, here are several questions that maybe useful for the authors if they decide to revise the paper:
1. Why solving the minimization problem in (6) is better than solving (2a) needs some clarifications? Can you provide numerical evidence in terms of accuracy and efficiency? The equation numbers here correspond to the equations labelled in the comments above.
2. Are there specific types of PDEs or solution structures for which the proposed tensorial kernel approach might be particularly well-suited?
3. Can you provide any theoretical insights or empirical evidence demonstrating advantages of this tensorial approach in terms of solution quality or generalization ability?
4. Can you clarify what DAKS is and what its relationship to their current work?
5. Can you provide additional state-of-the-art methods beyond PINN and finite-difference schemes?

---

> ### Author Response · Authors · 2024-11-22
> **Response to vnHZ**
>
> We thank the reviewer for their time. However, the reviewer seems to completely misunderstand the motivation and contribution of our work. The reviewer is even unclear what the major baseline method DAKS (Chen et al., 2021a) is, which directly motivates our work, and has been explained and highlighted in numerous places in our paper, e.g., Line 34-45 and whole Section 2.
>
> >C1: I just cannot see any new idea with this formulation. BTW, if the PDE is nonlinear, I believe RBF will result in a minimization problem as well
>
> R1: **It appears the reviewer has misunderstood our motivation and contribution, resulting in a superficial assessment**. The objective of our work is **not** to propose a new framework to replace kernel ridge regression. As highlighted throughout the paper, our aim is to develop an efficient kernel-based approach for PDE solving that (1) is computationally efficient and convenient to implement, and (2) provides convergence guarantees along with rate analysis. Evaluating a method without understanding its objectives is unproductive. Following the reviewer’s logic, the baseline method, DAKS (Chen et al., 2021a), could also be dismissed as merely a variation of kernel ridge regression, lacking "new ideas". By this reasoning, numerous neural network papers could similarly be deemed unoriginal just because they employ neural network formulations.
>
> While the reviewer proposed equations relevant to solving linear PDEs, our work focuses on developing a nonlinear PDE solver (see our test benchmarks). The context and results are entirely different, as nonlinear cases are much more challenging. For instance, in the nonlinear setting, it is generally not possible to achieve the optimal approximation in the interpolation form specified in Eq. (6) in the review comments.
>
> The real challenge is:  for **nonlinear** PDEs, if one applies kernel ridge regression directly, justifying **optimality and convergence** becomes very difficult. The reviewer might be aware of the the theoretical foundation of kernel ridge regression. When we solve an optimal recovery problem framed as minimizing $\left\lVert u\right\rVert_{RKHS}$ s.t. $u(z_j) = y_j (j=1 \ldots M)$, the representation theory guarantees that the optimal solution takes an interpolation form, $u(x) = \kappa(x, \mathbf{Z}) \mathbf{\alpha}$. Only with this form can we express the RKHS norm of $u$ is $\mathbf{\alpha} \mathbf{K} \mathbf{\alpha}$. Based on this result, the optimal recovery problem can be reformulated as a soft-constraint optimization problem, resulting in the well-known **kernel ridge regression**:
> $\min_\mathbf{\alpha} \frac{1}{M}\sum_j u(\mathbf{x}_j - y_j)^2 + \lambda\mathbf{\alpha}^\top \mathbf{K} \mathbf{\alpha} $.
>
> However, if you impose nonlinear constraints, such as $u^2(z_j) =y_j$, **the optimal solution no longer takes the simple linear form, and applying kernel ridge regression loses the foundation!**
>
> To address this, the prior work DAKS (Chen et al., 2021a) introduces a nested optimization formulation (see Section 2 of our paper for a detailed explanation). This approach ensures that at the inner level, the problem aligns with the standard optimal recovery form, allowing for a claim that, given derivative values $z^j_m$, the optimal solution for $u$ still takes the **standard linear/interpolation** form. However, this formulation requires augmenting the RKHS with derivative operators over the kernels, which inevitably introduces kernel derivative blocks in the kernel matrix.
>
> Our contribution, in contrast, is to maintain the favorable interpolation form without derivative augmentation (see Eq. (6) in our paper), allowing for a simpler implementation and more efficient computations while **reconstructing theoretical guarantees**!!. Achieving this required moving beyond the classical optimal recovery framework, as we constrain $u$ to a reduced search space within the RKHS. We have rigorously demonstrated that, even in this reduced space, convergence to the true PDE solution is achievable under mild regularity assumptions commonly employed in PDE convergence analysis. Moreover, our convergence rate remains comparable to that of DAKS. For the detailed proof, see Sections 4, A, and B, where we present a technical and highly nontrivial analysis.

---

> ### Author Response · Authors · 2024-11-22
> **Response to vnHZ**
>
> >C2: Beyond this issue, I also do not understand why the choice of tensorial kernel makes any sense except for computational convenience
>
> R2: As we have emphasized throughout the paper, our motivation for introducing the product kernel is rooted in reducing computational costs and enabling the use of a large number of collocation points --- strictly from a computational standpoint. We do **not** claim any additional implications beyond computational convenience. **It is surprising to see the reviewer’s strong focus on points that are tangential to our objectives, indicating a misunderstanding of our motivation**. Nevertheless, the product kernel corresponds to a tensor product structure in the latent feature space, consistent with the tensor product approach commonly employed in numerical methods [1].
>
> [1] ARNOLD, D. N., BOFFI, D., and BONIZZONI, F. (2012). Tensor product finite element differential
> forms and their approximation properties. arXiv preprint arXiv:1212.6559.
>
> >C3: Why solving the minimization problem in (6) is better than solving (2a) needs some clarifications? Can you provide numerical evidence in terms of accuracy and efficiency? The equation numbers here correspond to the equations labelled in the comments above.
>
> R3: We did try with direct optimization of the coefficients $\mathbf{\alpha}$ instead of using the approach in Eq. (6). However, we observed that this led to a solution error increased by 1 -- 2 orders of magnitude. Our empirical findings suggest that each coefficient $\mathbf{\alpha}$ globally influences the solution approximation, to which both the boundary condition fit and the PDE residuals are highly sensitive. As a result, small adjustments of the coefficients can cause significant perturbations, leading to an imbalance between these components.
> This makes the optimization process much more challenging than directly estimating the local solution values at the collocation points. Below, we present results from solving the Burgers' equation with $\nu=0.02$ to illustrate this effect.
>
> |                        	        | 1200     	| 2400     	| 4800     	|
> |------------------------|----------|----------|----------|
> | Our method             	| 5.40E-03 	| 7.83E-04 	| 3.21E-04 	|
> | Reviewer's sugguestion 	| 2.80E-02 	| 1.56E-03 	| 7.14E-04 	|

---

> ### Author Response · Authors · 2024-11-22
> **Response to vnHZ**
>
> >C4: Can you clarify what DAKS is and what its relationship to their current work?
>
> R4: As highlighted in Lines 353-355, DAKS (Chen et al., 2021a) is the prior work that motivates our approach. DAKS is based on the nested optimization framework introduced in Eq. (2) of Section 2. It incorporates a set of derivative values — more generally, the linear operators within the PDE, evaluated at the collocation points, denoted as $z^j_m$ in (2). DAKS then uses kernel interpolation to construct the solution approximation, as shown in Eq. (3), where the Gram matrix consists of sub-blocks computed by applying linear operators to the kernels (see Eqs. (4) and (5)). **All relevant details have been provided in Section 2**

---

> ### Comment · Reviewer_vnHZ · 2024-11-25
>
> I admitted that I am very confused reading this paper since the presentation does not help improving my understanding either. While the presentation states that the proposed work is trying to improve upon DAKS, but then it is very unclear to me why one wants to regress to a set of features that include C_{ij} corresponds to derivatives of the kernel as in (3)-(5) in the manuscript. Effectively, the PDE constraints readily provide matrix $A = K''+\alpha K$. At the same time, such a question does not seem to be fair since I am not reviewing DAKS. Second, the proposed new algorithm in this paper now seems to abandon the needs of regressing over C_{ij}, which is how it should be done in the first place. But then I am bothered with why would one devise an algorithm to beat previous method that is unclear. This stems me to look at what is actually being proposed if one solves a simple linear PDE, which helps me clarify what this paper seems to be about.
>
> Responding to C2, I am still not convinced why one should do the tensorial kernels. There must be cases when such idea is advantageous and/or not advantageous beyond computational costs. I understand your motivation but the current answer seems to be based on empirical evident of a few examples.
>
> Responding to C3, this comparison is not complete. The fact that minimizing over $\eta$ is better than $\alpha$ requires a lot of explanation. There are many factors that can affect the solutions (including optimization scheme, initial conditions, etc) and whether this phenomenon occurs only on 1 example is unclear.

---

### Official Review · Reviewer_JusD · 2024-11-03

**Soundness:** 3
**Presentation:** 4
**Contribution:** 2
**Rating:** 5
**Confidence:** 3

**Summary:**

This paper introduces a kernel learning framework for solving nonlinear partial differential equations (PDEs). Unlike a previous kernel solver (Chen et al., 2021a) that embeds differential operators within the kernel matrix, this new approach directly applies the operator to the kernel function, resulting in a hybrid method between traditional physics-informed neural networks (PINNs) and kernel methods. Additionally, the authors propose placing collocation points on a grid and utilizing a product kernel, so that the Gram matrix can be decomposed as a Kronecker product, significantly reducing computational costs.

**Strengths:**

The paper is well presented. The literature review seems adequate and I especially appreciated the introduction to the previous approach by (Chen et al., 2021a). Furthermore, I found particularly clever the idea of placing collocation points on a grid and utilizing a product kernel, so that the kernel decomposes into a Kronecker product.

**Weaknesses:**

I did not like the trade-off of making the kernel just a constant smaller (for instance, 1/3 smaller for the Burger's equation) at the cost of having to do an optimization similar to how regular PINNs are trained. Of course a 27 (3x3x3) times speed-up is not negligible, but with the authors approach, we lose the ability to do everything with fast Linear Algebra operations, and still do not get the advantages of training regular PINNs, where the colocation points can be randomly positioned and using stochastic gradients allows for using fewer collocation points per epoch. The product kernel trick was a nice idea, but I am convinced that it can also be applied in Chen et al., 2021a.

I am also skeptical of the timings reported by the authors in the appendix. In the experiments setup they say SKS is run using the ADAM optimizer for 1e6 epochs, but in the table 9 in the appendix, where they report time per iteration and total, the ratio is not 1e6. I also found it very surprising that the time per iteration is so small, especially compared to those of PINNs: I would think evaluating the derivative of the PINNs should be faster than evaluating this derivatives on the kernel version, even with the product trick.

**Questions:**

- Why learn $\eta$, instead of $K_{MM}^{-1} \eta$? Both are vectors of the same dimension, so it should not make a diference. Evaluating the RHKS regularization term is also possible by calculating $(K_{MM}^{-1} \eta)^T K_{MM} K_{MM}^{-1} \eta$.
- In the numerical experiments, the collocation points should be the same for SKS and DAKS, as I am convinced the higher errors of DAKS arise from random sampling: with random sampling, there may be "holes" in the domain without collocation points, where the error is bigger.

---

> ### Author Response · Authors · 2024-11-22
> **Response to JusD**
>
> We thank reviewer for their time, and we would like to reply with the following.
>
> > W1: I did not like the trade-off of making the kernel just a constant smaller. ... The product kernel trick was a nice idea, but I am convinced that it can also be applied in Chen et al., 2021a.
>
> In Chen et al., although each block of the full Gram matrix can be converted in kronecker
> product if product kernel is used. However, the full Gram matrix cannot be decomposed into
> kronecker product. Thus, we break the big Gram matrix into smaller ones and exploit the kronecker
> structure for efficient computation.
>
> > W2: In the experiments setup they say SKS is run using the ADAM optimizer for 1e6 epochs, but in the table 9 in the appendix, where they report time per iteration and total, the ratio is not 1e6.
>
> For reported runtime, total time is calculated with early stopping criterion. We stopped the optimization, if the performance
> stopped improving for 1000 iterations. This is the reason why our method total time is not 1e6 * per-iter time.
>
> > W3:  I also found it very surprising that the time per iteration is so small, especially compared to those of PINNs
>
> We believe our method time per iteration is surprisingly small compared to PINNs is due to the following:
> 1. Our method is based on linear operations, where we did not calculate gradient explicitly. But PINNs are based on
> autograd.
> 2. Optimization in our method is simpler and more convenient for modern libraries. Our method only needs to optimize
> parameter \eta, where PINNs need to optimize with computation graphs on derivative terms.
> 3. Our method is non-parametric, the only tunable parameters are the lengthscale parameters. We further improved our
> computational efficiency by fixing the kernel matrix terms at first iteration.
>
>
> >Q1: Why learn $\eta$, instead of $K^{-1}\eta$?
>
> Great question, we actually tried this approach! But the performance is just subpar, we believe it is due to $K^{-1}\eta$
> is hard to optimize. Below, we present results from solving the Burgers' equation with $\nu=0.02$ to illustrate this effect.
>
> |                        	| 1200     	| 2400     	| 4800     	|
> |------------------------|----------|----------|----------|
> | Our method             	| 5.40E-03 	| 7.83E-04 	| 3.21E-04 	|
> | Reviewer's sugguestion 	| 2.80E-02 	| 1.56E-03 	| 7.14E-04 	|
>
> > Q2: In the numerical experiments, the collocation points should be the same for SKS and DAKS, as I am convinced the higher errors of DAKS arise from random sampling: with random sampling, there may be "holes" in the domain without collocation points, where the error is bigger.
>
> We actually tested with both random sampling and grid sampling and random sampling is constant better than grid sampling,
> thus, we used 5 different seeds and reported the average.
>
> Below, we present results of burgers $\nu=0.02$ with two difference sampling methods.
>
> | Sampling method 	| 600      	| 1200     	| 2400     	| 4800     	|
> |-----------------|----------|----------|----------|----------|
> | Grid            	| 4.09E-01 	| 3.85E-01 	| 4.27E-02 	| 5.67E-02 	|
> | Random          	| 1.75E-02 	| 7.90E-03 	| 8.65E-04 	| 9.76E-05 	|

---

### Official Review · Reviewer_bwZk · 2024-11-05

**Soundness:** 2
**Presentation:** 2
**Contribution:** 2
**Rating:** 5
**Confidence:** 3

**Summary:**

The author proposes a very interesting kernel-based numerical solvers for solving nonlinear PDE.

**Strengths:**

The topic looks very interesting

**Weaknesses:**

1. The manuscript has not been well-written, so that the reader cannot find their motivation clearly.
2. The theoretical part has been shown rigorously.
3. The experiment part: description not clear

**Questions:**

If the authors can rewrite the manuscript more clearly, I would like change the grade.

---

### Official Review · Reviewer_Krce · 2024-11-08

**Soundness:** 2
**Presentation:** 4
**Contribution:** 4
**Rating:** 5
**Confidence:** 5

**Summary:**

The paper introduces and studies a novel kernel-based method for the approximation of the solution of a broad class of non-linear PDEs. The authors discuss computational aspects and provide error estimates in Sobolev spaces of appropriate regularity.
The method is a modification (actually, a significant simplification) of an existing method, but the novelty is relevant as it provides a much more efficient solution method, and proves corresponding modified theoretical guarantees on the error.

**Strengths:**

- The topic is of current interest, and the results are clearly presented and discussed.
- The new method is computationally efficient and rather simple, when compared to other related approaches.
- A throughout theoretical analysis is provided, even if one step needs clarification.

**Weaknesses:**

- The paper lacks a sufficient discussion of the existing work, especially as it does not acknowledge a large body of literature on kernel-based (symmetric and non symmetric) collocation methods. In particular the last paragraph of Section 5 (Related Work) states that the novelty over existing methods is the use of nodal values instead of coefficients, and in the use of an optimization problem instead of the solution of a linear system. Both these aspects are treated in the literature, sometimes also for nonlinear problem, see e.g. [1-5], even if certainly not for such general problems as (1). I would suggest to expand the discussion of existing work, and also clearly articulate how the approach differs from or improves upon these existing methods.



- The relation between Assumption 4.1 and the existence and uniqueness of a strong solution of (1) is unclear. This relation would be needed to interpret Lemma 4.2 and Proposition 4.3 (i.e., given a PDE (1), which $k$, $t$, $\tau$ should one expect in (14)).
This fact is reflected also in the numerical experiments: It's unclear if these examples fit into the assumptions of the theoretical results (having strong solutions of sufficient regularity), and if so, with which values of $k$, $t$, $\tau$. In more general terms, the numerical experiments should address the expected convergence rates, other than just errors with fixed discretisations.

- I'm not convinced by the argument leading to (32) in the proof of Lemma 4.2 in Appendix A. There is usually an issue in using a sampling inequality in this way. Namely, one has a domain $\mathcal M$ (in this case $\mathcal M=\mathcal T_i$), which has a corresponding critical value $h_0$ (depending on $\mathcal T_i$ via its diameter and boundary). Then, taking sufficiently many points in $\mathcal M$ one can have $h_i<h_0$ and thus apply a sampling inequality like the one of Proposition A.1 in (Batlle et al., 2023). Here however the only collocation point in $\mathcal T_i$ is $x_i$, and $h_i$ can not be made small without changing $\mathcal T_i$ itself, and thus $h_0$. Some oversampling inside $\mathcal T_i$ is usually needed. I suggest to clarify this point.

[1] K. Böhmer, R. Schaback, A Nonlinear Discretization Theory for Meshfree Collocation Methods applied to Quasilinear Elliptic Equations, ZAMM (2020)

[2] V. Bayona et al.,  RBF-FD formulas and convergence properties, Journal of Computational Physics (2010)

[3] I. Tominec, Residual Viscosity Stabilized RBF-FD Methods for Solving Nonlinear Conservation Laws, J Sci. Comp. (2022)

[4] Ka Chun Cheung et al, H^2-Convergence of Least-Squares Kernel Collocation Methods, SINUM (2018)

[5] N. Flyer et al., A guide to RBF-generated finite differences for nonlinear transport: Shallow water simulations on a sphere, J. Comp. Physics (2012)

**Questions:**

Apart from the points discussed above, there are the following minor points:
- Around and after (2), there is some notational confusion between $z^j_m$ (a value) and $z^j$ (a function, later evaluated at $x_m$).
- Eq. (23) in Appendix A: The norm in the lhs should be over $\Omega$, not $\mathcal X$. Moreover $u_M$ should be $u_M^*$.
- Eq. (26), first row: Arguments $x_m$ are missing in the lhs.
- Starting from (7) and in (many of) the following occurrences, the index in the second sum starts from $m=M_+1$, instead of $m=M_{\Omega}+1$.
- Use of sampling inequalities: I don't understand why the one of (Batlle et al., 2023) is needed in eq. (21) and later, and not a sampling inequality on the flat Omega (as e.g. [6]).

[6] H. Wendland and C. Rieger, Approximate Interpolation with Applications to Selecting Smoothing Parameters, Numerische Mathematik (2005)

---

### Meta-Review · Area_Chair_TpSn · 2024-12-19

**Metareview:**

The paper introduces and studies a novel kernel-based method for the approximation of the solution of a broad class of non-linear PDEs. The authors discuss computational aspects and provide error estimates in Sobolev spaces of appropriate regularity. Unfortunately, two reviewers argue that the proposed methodology does not introduce fundamentally new ideas. The formulations presented appear to replicate well-established methods, particularly in the context of radial basis functions. One of these two reviewers is more positive, acknowledging that the method is a modification (actually, a significant simplification) of an existing method, but the novelty is relevant as it provides a much more efficient solution method, and proves corresponding modified theoretical guarantees on the error. Nevertheless, they point out that the paper lacks a sufficient discussion of the existing work, especially as it does not acknowledge a large body of literature on kernel-based (symmetric and non symmetric) collocation methods. Finally, all reviewers have serious concerns about the experiments.

**Additional Comments On Reviewer Discussion:**

The reviewers raised numerous concerns about the novelty and effectiveness of the proposed method. There were some concerns about how the experiments were performed, and I do not feel they have been adequately addressed in the rebuttal.

---

### Decision · Program_Chairs · 2025-01-22

Reject